# Evaluation of a generalized knowledge-based planning performance for VMAT irradiation of breast and locoregional lymph nodes— Internal mammary and/or supraclavicular regions

**Maria Rago[1], Lorenzo Placidi[1,2]\*, Mattia Polsoni[3,4], Giulia Rambaldi[3,4], Davide Cusumano[2], Francesca Greco[2], Luca Indovina[2], Sebastiano Menna[2], Elisa Placidi[2], Gerardina Stimato[2], Stefania Teodoli[2], Gian Carlo Mattiucci[2], Silvia Chiesa[2], Fabio Marazzi[2], Valeria Masiello[2], Vincenzo Valentini[1,2], Marco De Spirito[1,2], Luigi Azario[1,2]**

**1** Università Cattolica del Sacro Cuore, Rome, Italy, **2** Fondazione Policlinico Universitario A. Gemelli IRCCS, Rome, Italy, **3** Fatebenefratelli Isola Tiberina, Ospedale San Giovanni Calibita, Rome, Italy, **4** Amethyst Radioterapia Italia, Isola Tiberina, Rome, Italy

\* lorenzo.placidi@policlinicogemelli.it

## Abstract

### Purpose

To evaluate the performance of eleven Knowledge-Based (KB) models for planning optimization (RapidPlan^tm (RP), Varian) of Volumetric Modulated Arc Therapy (VMAT) applied to whole breast comprehensive of nodal stations, internal mammary and/or supraclavicular regions.

### Methods and materials

Six RP models have been generated and trained based on 120 VMAT plans data set with different criteria. Two extra-structures were delineated: a PTV for the optimization and a ring structure. Five more models, twins of the previous models, have been created without the need of these structures.

### Results

All models were successfully validated on an independent cohort of 40 patients, 30 from the same institute that provided the training patients and 10 from an additional institute, with the resulting plans being of equal or better quality compared with the clinical plans. The internal validation shows that the models reduce the heart maximum dose of about 2 Gy, the mean dose of about 1 Gy and the $V_{20Gy}$ of 1.5 Gy on average. Model R and L together with model B without optimization structures ensured the best outcomes in the 20% of the values compared to other models. The external validation observed an average improvement of at least 16% for the $V_{5Gy}$ of lungs in RP plans. The mean heart dose and for the $V_{20Gy}$ for lung IPSI

**Data Availability Statement:** All relevant data are within the manuscript and its Supporting Information files.

**Funding:** The authors received no specific funding for this work.

**Competing interests:** The authors have declared that no competing interests exist.

were almost halved. The models reduce the maximum dose for the spinal canal of more than 2 Gy on average

## Conclusions

All KB models allow a homogeneous plan quality and some dosimetric gains, as we saw in both internal and external validation. Sub-KB models, developed by splitting right and left breast cases or including only whole breast with locoregional lymph nodes, have shown good performances, comparable but slightly worse than the general model. Finally, models generated without the optimization structures, performed better than the original ones.

## Introduction

The increasing complexity of radiotherapy treatment planning, mainly caused by the difficulty in sparing individual organs-at-risk (OARs), leads to a challenge to efficiently produce consistent, high-quality radiotherapy treatment plans [1, 2]. External beam radiotherapy plans require individually optimized planning. Plan optimization is very time consuming, mainly because several iterations in a trial-and error process have to be done before a clinically acceptable plan can be safely delivered to the patients for IMRT and VMAT treatments [3, 4].

Plan Quality Assurance (QA) in the treatment preparation workflow is an underestimated step since insufficient attention is given in evaluating whether a given plan can be improved. Plan quality [5–8] affects treatment outcomes in clinical trials, so it could be used as an instrument for QA. Although specific guidelines, which define the minimum standards for dose targets and OARs [9], can in a certain way smooth out the differences and the result planner-dependant plan quality, but they do not lead the planner to the optimal plan for the specific patient. This could be ascribed not to lack of skills or experience of the planner, but somewhat to his ability to focus upon a limited number of objectives at a time, because with several OARs in play, there may be a tendency to focus more on specific OARs and thereby disregarding the importance of other risk organs during the optimization process.

Automatic planning systems were recently developed, so that the inter-operator variability could be reduced, the planning time could be spared, and the plan's quality could possibly be improved [10–16]. The Knowledge based (KB) optimization approach, is a good alternative to the "automatic planning" because of its capability to improve both the plan consistency and the planning efficiency [17, 18].

KB models must be reviewed, refined, and validated through a complex and iterative process that requires great effort by the user [19–23], so their implementation may turn out to be time consuming, despite the promising results reported.

A KB approach takes implicitly in account the patient anatomy differences and could evaluate the entire DVH shape for a new patient based on the included DVHs in the model. Considering that, KB planning could help in finding sources of planning inconsistency across large facilities; for example, differences in physician OAR sparing preferences or weakness in staff training or communication [24].

RapidPlan[tm] (RP), the commercial KB software by Varian Medical Systems (Palo Alto, CA), allows a general improvement in the inter-patient consistency of the treatment plans, their intrinsic quality and the efficiency (time and workflow) of the process [25]. The RP approach harmonizes the practice among different centres [26–31] or among planners with different skills [32, 33] and could help the planner in achieving optimal dose distributions,

although in general the today's plan quality is of high level. The mechanical performance and the dosimetric accuracy of the RP, as well as the improvements in OAR sparing using RP planning were verified, showing that RP could be used in clinical practice [34].

The present study aimed to evaluate the performance of eleven KB models created with RP software for VMAT planning optimization applied to whole-breast irradiation comprehensive of nodal stations, Internal Mammary (IM) and/or SupraClavicular (SC) regions.

The breast is an elective choice as a site for KB planning investigation, but to the best of our knowledge, a generalized model for a breast target comprehensive of the locoregional lymph nodes has never been done for VMAT treatments. 3D Conformal Radiation Therapy (3DCRT) is still the most used technique for breast irradiation, but in some particular cases it is not sufficient to obtain a good target coverage, sparing at the same time the surrounding OARs. Moreover, 3DCRT plans can be very operator dependent. The patient groups who are characterized by particular anatomies or who have to treat the lymph nodes, especially the IM lymph nodes, might be most beneficial from the VMAT technique [35], in the view of dilemma in ensuring IM lymph nodes coverage while limiting central lung depth and maximum heart depth with 3DCRT [36]. In fact, VMAT breast treatments provide good target coverage and organs at risk (OAR) sparing [37–46] but, even more when the majority of the near nodal stations are included, it is time consuming and it requires the delineation of ad hoc additional structures to improve conformity and decrease the dose to the surrounding structures, in particular lungs and heart. Moreover, it had to be considered the enormous variability between different patients and geometries, which could result in particularly challenging plans for patients presenting complex anatomies.

Previous studies have already investigated KB approach for breast cancer, mainly focusing on whole breast with Simultaneous Integrated Boost (SIB) to the tumour bed [47, 48]. Another study conducted by van Duren-Koopman et al. demonstrated clinically competitive and efficient optimization with RP of hybrid VMAT in tangential chest-wall irradiation plus SC nodes [49]. As a step further, there is still a lack of information whether RP can provide promising dose solution for VMAT treatment of breast cancer with IM nodes involvement.

The goal of this study is to evaluate the feasibility of several RP models for whole breast irradiation and locoregional nodes and then to validate the models on internal and external patients' cohort, in order to appraise the robustness and flexibility of RP models. The end point is to have models that successfully and efficiently produce clinically acceptable plans for breast site within the departmental protocol and outside it, to understand which model performs better in every case.

## Methods and materials

A set of clinical plans elaborated from January 2017 to September 2019 was included in this retrospective study. Criteria for selection were breast cancer adjuvant radiotherapy; VMAT technique; IMN delineation; no indication to only-nodes volume radiotherapy (inclusion of also breast gland or chest wall). The geometry and dosimetry of the structure set of each plan are then parameterized and extracted in the models. Each model was built on a range of plans from 52 to 120 (depending on the model).

Then the training phase begins and once the training of the model is completed, the model must be evaluated. The software integrated statistics identified the possible outliers in the regression of the principal components, according to Cook's distance, a measure of the influence of individual training set cases on regression coefficients and eventually other statistic parameters like the studentized residual.

## Model configuration

Patients who received VMAT treatments to breast sites were retrospectively selected by searching in our institution database.

The breast Clinical Target Volume (CTV) is defined as the entire mammary gland, the CTV_surgical bed, if there is one, is defined as 1 cm around the surgical clips placed in the lumpectomy area, the supraclavicular nodes are defined as the CTV_SC and the internal mammary nodes are defined as the CTV_IM LN, its extension depends on the prescription. The Planning Target Volumes (PTV) are defined by adding an anisotropic margin: 2mm in the medio-lateral direction, 5mm in the antero-posterior direction and 10 mm in the cranio-caudal direction [50]. All the PTVs were cropped 3mm inside the body outline to exclude the skin and, for SIB cases, the sum of other PTVs is subtracted at the isotropic expansion of the CTV_Boost, named PTV_boost. A total dose ranged in 57.5–63.22 Gy in 25/29 fractions was prescribed to the boost volume (PTV_boost), and simultaneously 50–52.2 Gy to the whole breast PTV or whole breast with nodal stations. If a single volume had indication, dose prescription could be 40.05 Gy in 15 fractions or 50 Gy in 25 fractions. In detail, breast treatments included in the models are:

1. Whole Breast (WB): 40.05 or 50 Gy, respectively in 15 or 25 fractions

2. WB with Surged Bed Boost (WB+SB): 50/57.5 Gy, 50/60 Gy and 50/62.5 Gy in 25 fractions

3. WB with SC Lymph Nodes (WB+SC LN): 50 Gy in 25 fractions

4. WB with SC Lymph Nodes with Boost (WB+SC LN+SB): 50/57.5 Gy, 50/60 Gy and 50/62.5 Gy in 25 fractions

5. WB with SC and IM Lymph Nodes (WB+SC LN+IM LN): 50 Gy in 25 fractions

6. WB with SC and IM Lymph Nodes with Boost (WB+SC LN+IM LN+SB): 50/57.5 Gy, 50/60 Gy, 50/62.5 Gy in 25 fractions and 52.2/63.22 Gy in 29 fractions

All plans were delivered in our department and therefore approved by a radiation oncologist. The models were trained with selected plans to include a wide range of cases representative of our clinical practice. All patient data were anonymized.

A potential critical point in an automated process, is the use of the same (or not) optimization and calculation algorithms for generating the plans used to feed the model during the validation phase, as well as for the implementation in the clinical practice. In the present work, the clinical plans were generated with the PRO optimization engine and the AAA algorithm, while in the RP validation the PO optimization (PO version 13.6.23, Varian Medical Systems, Inc., Palo Alto, CA, USA) and Acuros dose calculation algorithms (version 13.2.63, Varian Medical Systems, Inc., Palo Alto, CA, USA) were used. PO was found to overcome PRO limitations for VMAT planning [51, 52] and Acuros is more accurate than AAA [53, 54]. Regarding the optimizer, the initial clinical plans might have been better if optimized using the duo PO-Acuros. To exclude this possibility and also to avoid the eventuality to ascribe the improved quality of the RP generated plan to the different optimizer, although the algorithms differences should in principle have no real impact in the use of RP, we recalculated the original plans with PO and Acuros before including them in the models. In this way, all the comparisons were between plans consistently generated by the PO optimizer and computed with Acuros, that was applied as the algorithm for the final dose calculation as well as for the intermediate dose calculation.

Approaches such as Deep Inspiration breath-hold (DBHI) techniques during VMAT irradiation are suggested in literature [55–57] in order to avoid significant variations in dosage to the PTV and to reduce the dose delivered to the heart and lungs volumes, in particular for left

side breast cancer. However, in our data set only two patients were treated with this technique, mainly because a very good compliance of the patient is necessary to ensure an optimal delivery.

VMAT plans were optimized for 6MV photon beams with two or three partial arcs, collimator angle of 20–30°/330-340° and the isocenter opportunely placed in the target. Additional partial arcs were added in some more challenging cases, always within the limits of anterior/oblique to posterior incidence. All plans were normalized to the mean dose to PTV as for institutional policy in clinical routine and in compliance with the ICRU recommendations. The Acuros-XB dose calculation algorithm was adopted with a dose grid resolution of 2.5mm, as well as the slice thickness used for the CT (GE Optima 660) image on which the dose is calculated is 2.5 mm.

One hundred and twenty VMAT plans (60 left-sided, 60 right-sided breasts), 62 of them with SIB, were selected for the training of the DVH estimation models. These plans were manually performed by expert planners and approved by radiation oncologist based on the standard procedures of our department. All the plans selected for model training were checked for their quality before to their inclusion in the model, in terms of the maximum ($D_{max}$) and mean dose ($D_{mean}$) of the PTVs and OARs and in terms of the dose-volume parameters of PTVs and OARs as required by our clinical protocol (Quantec). The following parameters have been also calculated:

- ■. Homogeneity Index HI; defined as $\frac{D_{2\%}-D_{98\%}}{D_{50\%}}$, where $D_{98\%}$, $D_{2\%}$, and $D_{50\%}$ are the doses received by 98%, 2%, and 50% of the PTV respectively

- ■. Homogeneity Index $HI_{95}$; defined as $\frac{D_{5\%}}{D_{95\%}}$, where $D_{95\%}$, $D_{5\%}$ are the doses received by 95% and 5% of the PTV respectively

- ■. The 95% isodose Conformity Index $CI_{95}$; defined as $\frac{V_{95\%}}{V_{PTV}}$, where $V_{95\%}$ is the volume covered by 95% of the prescribed dose and $V_{PTV}$ is the PTV volume

- ■. The 100% isodose conformity index $CI_{100}$; defined as $\frac{V_{100\%}}{V_{PTV}}$,), where $V_{100\%}$ is the volume covered by 100% of the prescribed dose and the $V_{PTV}$ as previously described

Six different RP models have been generated and trained based on 120 VMAT plans data set:

1. Model B (120 VMAT plans) includes plans with whole right and left breast irradiation with locoregional lymph nodes.

2. Model LN is a subgroup of model B (100 VMAT plans) without plans with only whole breast irradiation.

3. Model IM LN is a subgroup of model LN (52 VMAT plans) with only plans that include among the lymph nodes, the internal mammary nodes irradiation.

4. Model R is a subgroup of model B (60 VMAT plans) without left plans.

5. Model L is a subgroup of model B (60 VMAT plans) without the right plans.

6. Model SIB is a subgroup of model B (62 VMAT plans) with only plans with SIB irradiation.

For RP generated plans, only two extra-structures were delineated, already one step ahead of the 8 to 10 structures that are usually created in the reference planes. The first structure was the PTV enlarged with a margin of 1 mm, cropped from the Body by 3 mm and for SIB cases a Boolean difference is performed between the PTV_all and the PTV_boost with a margin of 3

mm. The second was a ring structure defined as expansion of the PTV cropped of 0.3 cm from the PTV edge and 3 cm thick, to ensure a good dose conformation together with the appropriate NTO choice (with a fall off of 0.7).

The other five models, twins of the previous models, except for SIB model, have been created so that the two aforementioned extra structures don't have to be contoured and included during the optimization phase. These models are below named like their own gemini with the wording "No OS", namely "without optimization structures".

The OARs included in the training phase are: ipsilateral and contralateral lungs, their sum structure as lungs, contralateral breast, heart, spinal canal, Left Anterior Descending Coronary Artery (LADCA), esophagus and thyroid. Upper, lower, mean and line optimization objectives and their priorities were created in the model configuration for target and OAR that aim to achieve the standard protocol objectives of the department. For the serial organs, where point maximum dose constraints were the only constraints in the departmental protocol, a fixed upper objective and priority was used. For heart and lungs, where the accepted dose was more influenced by geometric factors, the RP model was used to generate a line objective and priority.

Potentially incorrect optimization line objectives for the estimated structures could be provided because of some outliers not properly checked. Any contours that were highlighted as outliers in the RP statistics were individually assessed and removed from the model if they were assumed to be outliers. No whole plans were directly removed during the training process for the breast models, but if the number of outlier structures exceeded half of the total number of structures, the whole plan was therefore removed.

The choice of the proper objectives and priorities adopted to create a model is an additional important factor related to the model quality. The placement of the line objective below the lower boundary of the prediction DVH improves the average plan quality. The good results of the plans generated with RP could come from the combination of the two objectives included in the model: the generated line-objective and the mean objective, both with generated priorities. Line objectives of a specified OAR refer to the 'most-likely occurring' DVH curve within the estimated DVH range and correspond to the low edge of the DVH range (mean estimated DVH ±one standard deviation).

After accomplishing several fine-tuning tests by planning sample patients, the KB-based template for planning optimization was finally generated and used for automatic optimization.

## Model validation

The validation phase consists of using the trained models to estimate DVHs on a group of patients with similar characteristics compared to those used to train the models. A set of 40 plans, not used for training the models, were selected: 30 (12 left, 18 right) from our department (clinic 1) and 10 (all right) from another institute (clinic 2). All clinical plans were approved for use ("Reference" plans in the following) and re-optimized with the above detailed RP models.

Sharing the model between centers has been quite easy, since all the necessary data can be exported in binary encrypted format from one center and simply re-imported into the Eclipse planning system of the destination center. No exchange of any patient data has been necessary for the purpose. The affinity among the centers would imply reasonable similar practice, protocols and some homogeneity in the patients' population. No special conditions were imposed to the testing center to strictly adhere to the model definitions, in terms of contouring rules for example, but rather the aim of the study was to appraise the possibility to use the same model within a real world environment, mimicking routine practice in different institutes.

The DVHs of the clinical plans in the validation patients were compared with the estimated DVHs obtained from each model. During the RP based optimization, no changes of the objectives nor priorities were allowed to exclude any operator dependent bias.

Standard quantitative and qualitative assessment of the DVHs was performed by inspecting the above mentioned dose volume parameters for either the targets, aiming to coverage and homogeneity information, and for the OARs, aiming to meaningful metrics for organs sparing, reported in more detail in the tables that show the obtained results, to appraise the quality of the model-based optimized plans versus the clinically accepted baseline benchmark.

## Statistical analysis

Statistical analysis was performed to compare the different dosimetric parameters of RP plans and manual plans. The Shapiro-Wilk test (OriginPRO by OriginLab (version 8.1)) was performed to verify the normality of the data. For normally distributed data, paired t tests were used to compare the different parameters. For non-normally distributed data, the Wilcoxon signed rank test has been performed. The tests assumed a null hypothesis, and the difference was considered statistically significant at $p < 0.05$(**), and highly significant for $p < 0.01$(***), but also a tendency to significance is pointed out if $p < 0.1$(*).

## Results

### Internal validation

The models were validated on an independent cohort of 30 patients of our department, with the resulting plans being significantly faster and of equal or better quality compared with the clinical plans.

The model training statistics given by the system showed acceptable model fit with, among the other parameters, an average chi-square (Pearson) for the regression model parameters of 1.06±0.12 for model B, of 1.03±0.03 for model LN, of 1.07±0.07 for model IM LN, of 1.05±0.03 for model R, of 1.08±0.04 for model L and of 1.07±0.09 for model SIB. The same average chi-square is for twin models without optimization structures.

When a model is applied to a new patient, the user is alerted about the structures that are flagged as outliers, i.e. presenting features different from the same parameters of the plans in the model. Evaluating the OARs for the 30 validation patients from clinic 1, a range of [1.7–3.0]% were flagged as outliers depending on the chosen model (red outliers), [2.4–4.1]% as outliers outwith the 90˚ percentile or beneath the 10˚ percentile but still under the maximum value or over the minimum value of the box plots (yellow outliers). Evaluating the parameters that the RP engine takes into account in the creation of the plan's objectives, for the volume parameters a range of [0.4–5.2]% were flagged as outliers depending on the chosen model, [0–6.3] % as outliers outwith the 90˚ percentile or beneath the 10˚ percentile; for the PCS a range of [0.4–5.2]% were flagged as outliers depending on the chosen model, [0–5.2]% as outliers outwith the 90˚ percentile or beneath the 10˚ percentile.

It is interesting to discover that almost the totality of these outliers turned to good value for the specific OAR, or the value of the corresponding OAR fell in the right range of constraints after the optimization phase ("green value"). In detail, a range [66.7–100]% of the red outliers, depending on the model, turned to be a "green value" after the optimization for all the parameters of the OARs and the 100% of the yellow outliers turned into a "green value". For the SIB model only, 16.7% of the yellow outliers leads after the optimization to some parameters for the specific OAR that do not fall within the optimal dosimetric range. More details are included in S4 and S5 Tables in S1 File.

**Table 1. Overview and comparison of relevant DVH parameters for OARs, averaged for the 30 patients of validation set, both for reference and model-based plans.**

| Model | Reference | | B | | | B_No OS | | | LN | | | LN_No OS | | |
|---|---|---|---|---|---|---|---|---|---|---|---|---|---|---|
| OAR | Average | Dev_st | Average | Dev_st | p value | Average | Dev_st | p value | Average | Dev_st | p value | Average | Dev_st | p value |
| **Breast CNTR** | | | | | | | | | | | | | | |
| Dmax (Gy) | 14.0 | ± 3.7 | 14.9 | ± 2.5 | | *14.2* | *± 2.6* | (*) | 14.6 | ± 2.5 | | 14.0 | ± 2.5 | |
| Dmean (Gy) | 4.3 | ± 1.1 | 4.7 | ± 1.0 | | *4.5* | *± 0.9* | (***) | *4.7* | *± 1.0* | (***) | 4.5 | ± 0.9 | |
| V$_{10Gy}$<5% | 2.4 | ± 2.0 | 3.2 | ± 2.3 | | *2.3* | *± 2.1* | (*) | *3.1* | *± 2.3* | (**) | 2.1 | ± 1.8 | |
| **Lungs** | | | | | | | | | | | | | | |
| Dmax (Gy) | 51.8 | ± 5.0 | 51.6 | ± 4.9 | | 51.7 | ± 5.0 | | *51.3* | *± 5.0* | (**) | 51.7 | ± 5.4 | |
| Dmean (Gy) | 9.4 | ± 2.0 | *9.3* | *± 1.6* | (*) | 9.1 | ± 1.7 | | 9.5 | ± 1.6 | | 9.3 | ± 1.7 | |
| V$_{5Gy}$<60% | 51.2 | ± 9.4 | *51.1* | *± 7.9* | (***) | 48.9 | ± 8.5 | | 52.1 | ± 7.5 | | 50.3 | ± 7.6 | |
| **Lung IPSI** | | | | | | | | | | | | | | |
| Dmax (Gy) | 51.8 | ± 5.0 | 51.6 | ± 4.9 | | 51.8 | ± 5.0 | | *51.3* | *± 5.0* | (**) | 51.7 | ± 5.4 | |
| Dmean (Gy) | 14.7 | ± 3.2 | 14.7 | ± 2.4 | | 14.4 | ± 2.8 | | 15.0 | ± 2.5 | | 14.7 | ± 2.7 | |
| V$_{20Gy}$<40% | 26.2 | ± 8.6 | 26.4 | ± 7.2 | | 26.1 | ± 7.2 | | 27.1 | ± 7.1 | | 26.9 | ± 7.4 | |
| **Lung CNTR** | | | | | | | | | | | | | | |
| Dmax (Gy) | 21.7 | ± 9.2 | 22.1 | ± 8.5 | | 21.1 | ± 8.1 | | *22.6* | *8.6* | (*) | 21.6 | ± 7.9 | |
| Dmean (Gy) | 3.8 | ± 1.0 | *3.8* | *± 1.0* | (**) | 3.6 | ± 0.9 | | 3.9 | 1.0 | | *3.7* | *± 0.9* | (**) |
| V$_{10Gy}$<5% | 4.5 | ± 4.0 | 4.9 | ± 3.9 | | 4.2 | ± 3.7 | | 5.4 | 4.1 | | 4.5 | ± 3.8 | |
| **Spinal canal** | | | | | | | | | | | | | | |
| Dmax (Gy) | 18.0 | ± 5.1 | 17.3 | ± 3.3 | | 16.8 | 3.3 | | 17.5 | 3.6 | | 16.9 | 3.3 | |
| Dmean (Gy) | 3.7 | ± 1.3 | 3.7 | ± 1.1 | | 3.6 | 1.1 | | 3.7 | 1.2 | | 3.6 | 1.2 | |
| **Heart** | | | | | | | | | | | | | | |
| Dmax (Gy) | 31.2 | ± 13.8 | *28.9* | *± 13.3* | (***) | *28.1* | *± 13.7* | (***) | *29.3* | *± 13.0* | (***) | *28.4* | *± 13.4* | (***) |
| Dmean (Gy) | 5.8 | ± 2.4 | *5.0* | *± 1.5* | (***) | *4.7* | *± 1.5* | (***) | *5.0* | *± 1.6* | (***) | *4.8* | *± 1.5* | (***) |
| V$_{20Gy}$<10% | 3.4 | ± 4.3 | *2.0* | *± 2.2* | (**) | *1.8* | *± 2.1* | (**) | *2.0* | *± 2.3* | (**) | *1.8* | *± 2.3* | (**) |
| **LADCA** | | | | | | | | | | | | | | |
| Dmax (Gy) | 18.1 | ± 8.5 | *17.2* | *± 7.5* | (**) | *16.4* | *± 6.7* | (**) | 17.2 | ± 7.5 | | 17.1 | ± 7.5 | |
| Dmean (Gy) | 9.5 | ± 3.7 | 10.4 | ± 4.3 | | *9.8* | *± 3.8* | (**) | *10.5* | *± 4.4* | (*) | 10.2 | ± 4.4 | |
| V$_{20Gy}$<10% | 5.2 | ± 6.7 | 10.5 | ± 17.1 | | 11.1 | ± 19.2 | | 7.0 | ± 9.1 | | 6.5 | ± 7.9 | |
| **Esophagus** | | | | | | | | | | | | | | |
| Dmax (Gy) | 37.8 | ± 11.4 | 37.7 | ± 11.4 | | 37.2 | ± 11.9 | | 37.7 | ± 11.4 | | 37.7 | ± 11.4 | |
| **Thyroid** | | | | | | | | | | | | | | |
| V$_{40Gy}$<20% | 14.2 | ± 16.8 | 15.8 | ± 16.6 | | 15.4 | ± 16.3 | | 15.7 | ± 16.4 | | 15.8 | ± 16.8 | |

P value is reported with *, ** or *** when significant as explained in the text.

In general, all KB-based plans were clinically acceptable in terms of PTVs coverage and OAR sparing. The PTV/OARs average dose-volume objectives were used to appraise the quality of the reference and RP dose distributions and were quantitatively analysed for PTV and OAR to investigate the differences. All DVH parameters are listed in detail in Tables 1–4 and S1–S4 Tables in S1 File. Average DVHs of PTVs and OARs for RP plans were compared to the reference plans and are shown in Fig 1 and in S1–S3 Figs.

About the 1% of 1141 analysed dose-volume objectives in the clinical plans failed to reach the optimal constraints, while the respective RP plans succeeded (more details are reported in S6 Table in S1 File). In 6 out of 30 evaluation data set plans, reduced to 2 or 0 cases out of 30 for models without the optimization structures, the RP plan failed at least one of the dose-volume constraints compared to the delivered plan, but with a mean difference of less than 6%. Vice versa, when RP brings the constraints below the optimal value, the average difference is up to 20%.

**Table 2. Overview and comparison of relevant DVH parameters for PTVs, averaged for the 30 patients of validation set, both for reference and model-based plans.**

| Model | Reference | | B | | | B_No OS | | | LN | | | LN_No OS | | |
|---|---|---|---|---|---|---|---|---|---|---|---|---|---|---|
| PTV | Average | Dev_st | Average | Dev_st | p value | Average | Dev_st | p value | Average | Dev_st | p value | Average | Dev_st | p value |
| **PTV RA** | | | | | | | | | | | | | | |
| **Dmax (Gy)** | 55.5 | ± 5.3 | 55.5 | ± 5.0 | | 55.3 | ± 5.1 | | 55.7 | ± 5.0 | | 55.2 | ± 5.2 | |
| $V_{95\%}>95\%$ | 97.5 | ± 2.4 | *97.0* | *± 2.4* | (***) | *96.6* | *± 2.0* | (**) | 97.1 | ± 2.3 | | *96.7* | *± 1.9* | (***) |
| $V_{105\%}<5\%$ | 4.3 | ± 5.5 | *3.7* | *± 4.1* | (**) | 3.4 | ± 4.1 | | 3.6 | ± 4.1 | | *3.1* | *± 3.9* | (***) |
| **PTV Surg Bed** | | | | | | | | | | | | | | |
| **Dmax (Gy)** | 62.4 | ± 2.7 | 62.2 | ± 2.7 | | 62.1 | ± 2.8 | | 62.2 | ± 2.7 | | 62.1 | ± 2.8 | |
| $V_{95\%}>95\%$ | 99.5 | ± 0.5 | 99.2 | ± 0.6 | | 99.4 | ± 0.5 | | *99.3* | *± 0.6* | (*) | 99.3 | ± 0.5 | |
| $V_{105\%}<5\%$ | 0.06 | ± 0.13 | 0.04 | ± 0.09 | | 0.10 | ± 0.21 | | 0.09 | ± 0.19 | | 0.04 | ± 0.1 | |
| **CI 100%** | | | | | | | | | | | | | | |
| $V_{isodose\ 100\%}/V_{PTV}$ | 0.57 | ± 0.06 | *0.58* | *± 0.06* | (***) | 0.59 | ± 0.05 | | *0.58* | *± 0.06* | (**) | *0.59* | *± 0.06* | (***) |
| **CI 95%** | | | | | | | | | | | | | | |
| $V_{isodose\ 95\%}/V_{PTV}$ | 1.10 | ± 0.07 | 1.08 | ± 0.07 | | *1.09* | *± 0.08* | (**) | *1.08* | *± 0.07* | (*) | 1.08 | ± 0.07 | |
| **HI** | | | | | | | | | | | | | | |
| $(D_{2\%}-D_{98\%})/D_{50\%}$ | 0.10 | ± 0.07 | 0.10 | ± 0.07 | | 0.10 | ± 0.07 | | 0.10 | ± 0.07 | | 0.09 | ± 0.07 | |
| **HI 5/95** | | | | | | | | | | | | | | |
| $D_{5\%}/D_{95\%}$ | 1.07 | ± 0.05 | 1.08 | ± 0.06 | | *1.07* | *± 0.06* | (**) | 1.07 | ± 0.06 | | 1.07 | ± 0.06 | |

P value is reported with *, ** or *** when significant as explained in the text.

The analysis showed that RP based optimizations lead to modest but systematic improvements in OAR sparing. Quantitative improvements were observed in RP plans, especially for heart and spinal canal doses. As shown in Tables 1 and 3 and S1 Table in S1 File, the models reduce the heart maximum dose of about 2 Gy, mean dose of about 1 Gy and the $V_{20}$ of 1.5 Gy on average. Moreover, the maximum dose for the spinal canal is about 1 Gy less on average, although not statistically significant. Small improvements are observed for the $V_{5Gy}$ and the $D_{mean}$ of lungs in RP plans, although they are significant only for the B model.

Some OARs shown slight dose increases, such as the contralateral breast or the LADCA, for LN and IM LN models. The same parameters turn to be slightly but statistically improved in model B, for example the $V_{10Gy}$ for the contralateral breast or the $D_{max}$ for the LADCA. Almost no statistically significant results were observed for lung IPSI, spinal canal, oesophagus or thyroid.

These improvements are confirmed in the models created without any other optimization structures, corroborating the general idea that RP allows a gain also in planning time and gives planners the possibility not to use additional optimization structures to better conform the dose in the target and to better spare the OARs.

Model B without optimization structures results in a better OAR sparing in the 30% of the values if compared to other models, Model R and L together with model B without optimization structures, considering both OAR sparing and PTV coverage, ensuing the best outcomes in about the 20% of the values compared to other models, as it can be seen in Fig 2.

## External validation

The above mentioned results are confirmed by the external validation, performed on 10 plans, all right side breasts, whose 9 were treated in SIB in 25 fractions with a dose prescription of 50/57.5 Gy in total (2/2.3 Gy per fraction) and 1 was a single breast volume without boost treated in 16 fraction for a total dose of 42.72 Gy (2.67 Gy per fraction).

**Table 3. Overview and comparison of relevant DVH parameters for OARs, averaged for the 18 patients of validation set, treated on the right side and for the 12 patients treated on the left side, both for reference and model-based plans.**

| Model | Reference R | | R | | | R_No OS | | | Reference L | | L | | | L_No OS | | |
|---|---|---|---|---|---|---|---|---|---|---|---|---|---|---|---|---|
| OAR | Average | Dev_st | Average | Dev_st | p value | Average | Dev_st | p value | Average | Dev_st | Average | Dev_st | p value | Average | Dev_st | p value |
| **Breast CNTR** | | | | | | | | | | | | | | | | |
| Dmax (Gy) | 13.1 | ± 3.5 | 14.8 | ± 2.2 | (**) | 14.0 | ± 2.4 | | 15.4 | ± 3.6 | 15.2 | ± 3.1 | | 14.7 | ± 2.5 | |
| Dmean (Gy) | 4.1 | ± 1.1 | 4.5 | ± 0.9 | (*) | 4.3 | ± 0.8 | | 4.7 | ± 1.0 | 5.0 | ± 1.0 | (**) | 4.7 | ± 1.0 | |
| $V_{10Gy}<5\%$ | 2.0 | ± 1.8 | 3.3 | ± 2.5 | (**) | 2.0 | ± 1.6 | | 3.0 | ± 2.4 | 2.8 | ± 2.4 | | 2.2 | ± 2.1 | |
| **Lungs** | | | | | | | | | | | | | | | | |
| Dmax (Gy) | 52.3 | ± 4.7 | 52.0 | ± 4.5 | | 52.0 | ± 4.9 | | 51.0 | ± 5.6 | 50.7 | ± 5.9 | | 50.8 | ± 6.4 | |
| Dmean (Gy) | 9.9 | ± 1.6 | 9.8 | ± 1.9 | | 9.5 | ± 2.0 | | 8.7 | ± 2.3 | 9.0 | ± 1.7 | (**) | 8.8 | ± 1.9 | |
| $V_{5Gy}<60\%$ | 52.2 | ± 8.3 | 52.4 | ± 10.1 | | 50.1 | ± 10.8 | (**) | 49.9 | ± 11.1 | 51.5 | ± 8.2 | (**) | 50.5 | ± 8.5 | |
| **Lung IPSI** | | | | | | | | | | | | | | | | |
| Dmax (Gy) | 52.3 | ± 4.7 | 52.1 | ± 4.5 | | 52.0 | ± 4.9 | | 51.0 | ± 5.6 | 50.7 | ± 5.9 | | 50.8 | ± 6.4 | |
| Dmean (Gy) | 15.3 | ± 2.4 | 15.1 | ± 3.1 | | 14.7 | ± 3.4 | | 14.0 | ± 4.1 | 14.3 | ± 2.7 | | 14.1 | ± 3.2 | |
| $V_{20Gy}<40\%$ | 27.4 | ± 6.6 | 27.0 | ± 8.0 | | 26.3 | ± 8.6 | (**) | 24.4 | ± 11.1 | 26.1 | ± 8.5 | (**) | 26.4 | ± 8.9 | |
| **Lung CNTR** | | | | | | | | | | | | | | | | |
| Dmax (Gy) | 17.6 | ± 4.9 | 18.7 | ± 5.0 | | 17.3 | ± 4.6 | | 27.9 | ± 10.8 | 28.3 | ± 10.0 | | 27.1 | ± 9.1 | |
| Dmean (Gy) | 3.3 | ± 0.9 | 3.5 | ± 0.6 | (*) | 3.3 | ± 0.6 | | 4.4 | ± 0.9 | 4.7 | ± 0.9 | | 4.6 | ± 0.8 | |
| $V_{10Gy}<5\%$ | 2.7 | ± 2.8 | 3.2 | ± 2.1 | | 2.3 | ± 1.9 | | 7.1 | ± 4.2 | 10.5 | ± 4.1 | (**) | 9.0 | ± 3.8 | (*) |
| **Heart** | | | | | | | | | | | | | | | | |
| Dmax (Gy) | 24.9 | ± 12.7 | 22.8 | ± 11.1 | (**) | 21.4 | ± 10.8 | (*) | 40.6 | ± 9.6 | 39.1 | ± 9.3 | (**) | 38.4 | ± 10.5 | (***) |
| Dmean (Gy) | 5.0 | ± 1.9 | 4.4 | ± 0.9 | (**) | 4.2 | ± 0.9 | (**) | 6.9 | ± 2.5 | 5.7 | ± 1.7 | (**) | 5.5 | ± 1.6 | (***) |
| $V_{20Gy}<10\%$ | 1.7 | ± 2.5 | 0.7 | ± 0.8 | (*) | 0.5 | ± 0.8 | | 4.7 | ± 5.1 | 2.7 | ± 2.5 | | 2.5 | ± 2.6 | |
| **LADCA** | | | | | | | | | | | | | | | | |
| Dmax (Gy) | 6.7 | ± 3.3 | 8.1 | ± 2.4 | | 7.7 | ± 2.2 | | 20.3 | ± 7.3 | 20.1 | ± 6.9 | | 19.5 | ± 6.6 | |
| Dmean (Gy) | 4.9 | ± 3.3 | 6.3 | ± 1.5 | | 6.2 | ± 1.7 | | 10.4 | ± 3.2 | 11.9 | ± 4.5 | (**) | 11.6 | ± 4.3 | (*) |
| $V_{20Gy}<10\%$ | | | | | | | | | 5.2 | ± 6.7 | 12.1 | ± 13.3 | (*) | 9.6 | ± 16.5 | |
| **Spinal canal** | | | | | | | | | | | | | | | | |
| Dmax (Gy) | 17.5 | ± 2.9 | 17.6 | ± 2.6 | | 17.0 | ± 2.1 | | 18.8 | ± 7.2 | 17.2 | ± 4.7 | | 16.7 | ± 4.4 | |
| Dmean (Gy) | 3.9 | ± 1.4 | 3.8 | ± 1.4 | | 3.7 | ± 1.3 | | 3.6 | ± 1.1 | 3.6 | ± 0.8 | | 3.4 | ± 0.9 | |
| **Esophagus** | | | | | | | | | | | | | | | | |
| Dmax (Gy) | 34.6 | ± 9.9 | 35.0 | ± 9.6 | | 34.7 | ± 10.1 | | 42.2 | ± 12.2 | 41.7 | ± 12.4 | | 41.5 | ± 12.4 | |
| **Thyroid** | | | | | | | | | | | | | | | | |
| $V_{40Gy}<20\%$ | 11.3 | ± 14.1 | 12.2 | ± 14.0 | | 12.0 | ± 13.4 | | 17.8 | ± 19.8 | 19.9 | ± 19.2 | | 19.6 | ± 19.1 | |

P value is reported with *, ** or *** when significant as explained in the text.

The treated volumes comprehend the whole right breast without lymph nodes, eventually with a boost volume around the surgical clips simultaneously integrated into the irradiation protocol. The contouring strategies were not changed, but the planning ones were slightly modified since the protocol followed by the medical physicists of the clinic 2 is not the QUANTEC one but the RTOG 1005 [58] and the VMAT constraints showed in Boman et al [59].

**Table 4. Overview and comparison of relevant DVH parameters for PTVs, averaged for the 18 patients of validation set, treated on the right side and for the 12 patients treated on the left side, both for reference and model-based plans.**

| Model | Reference R | | R | | | R_No OS | | | Reference L | | L | | | L_No OS | | |
|---|---|---|---|---|---|---|---|---|---|---|---|---|---|---|---|---|
| PTV | Average | Dev_st | Average | Dev_st | p value | Average | Dev_st | p value | Average | Dev_st | Average | Dev_st | p value | Average | Dev_st | p value |
| **PTV RA** | | | | | | | | | | | | | | | | |
| Dmax (Gy) | 55.4 | ± 4.5 | 55.8 | ± 4.5 | | 55.4 | ± 4.4 | | 55.6 | ± 6.5 | 55.4 | ± 6.1 | | 55.1 | ± 6.1 | |
| $V_{95\%} > 95\%$ | 97.0 | ± 2.8 | 96.8 | ± 3.0 | | 96.6 | ± 2.2 | | 98.2 | ± 1.2 | 97.5 | ± 1.5 | (*) | 97.0 | ± 1.6 | (**) |
| $V_{105\%} < 5\%$ | 3.0 | ± 4.4 | 3.3 | ± 4.1 | | 2.6 | ± 3.7 | | 6.2 | ± 6.7 | 4.7 | ± 4.5 | (**) | 3.8 | ± 3.7 | (**) |
| **PTV Surg Bed** | | | | | | | | | | | | | | | | |
| Dmax (Gy) | 62.9 | ± 3.5 | 63.2 | ± 3.5 | | 62.7 | ± 3.5 | | 62.1 | ± 2.3 | 61.7 | ± 2.1 | | 61.6 | ± 2.2 | |
| $V_{95\%} > 95\%$ | 99.5 | ± 0.6 | 99.0 | ± 0.7 | (*) | 99.5 | ± 0.4 | | 99.5 | ± 0.3 | 99.3 | ± 0.5 | | 99.3 | ± 0.5 | |
| $V_{105\%} < 5\%$ | 0.02 | ± 0.03 | 0.06 | ± 0.09 | | 0.04 | ± 0.07 | | 0.07 | ± 0.15 | 0.02 | ± 0.03 | | 0.01 | ± 0.01 | |
| **CI 100%** | | | | | | | | | | | | | | | | |
| $V_{isodose\ 100\%}/V_{PTV}$ | 0.59 | ± 0.07 | 0.58 | ± 0.07 | | 0.60 | ± 0.07 | | 0.54 | ± 0.03 | 0.57 | ± 0.03 | (***) | 0.58 | ± 0.03 | (***) |
| **CI 95%** | | | | | | | | | | | | | | | | |
| $V_{isodose\ 95\%}/V_{PTV}$ | 1.08 | ± 0.06 | 1.07 | ± 0.07 | | 1.09 | ± 0.09 | | 1.13 | ± 0.07 | 1.11 | ± 0.05 | (*) | 1.10 | ± 0.07 | |
| **HI** | | | | | | | | | | | | | | | | |
| $(D_{2\%}-D_{98\%})/D_{50\%}$ | 0.11 | ± 0.09 | 0.11 | ± 0.09 | | 0.10 | ± 0.09 | | 0.08 | ± 0.02 | 0.08 | ± 0.02 | | 0.08 | ± 0.02 | |
| **HI 5/95** | | | | | | | | | | | | | | | | |
| $D_{5\%}/D_{95\%}$ | 1.08 | ± 0.07 | 1.08 | ± 0.07 | | 1.08 | ± 0.07 | | 1.06 | ± 0.01 | 1.06 | ± 0.02 | | 1.06 | ± 0.02 | |

P value is reported with *, ** or *** when significant as explained in the text.

Evaluating the OARs for the 10 validation patients from clinic 2, a range of [21.4–48.6]% were flagged as outliers depending on the chosen model (red outliers), [0–17.1]% as outliers outwith the 90° percentile or beneath the 10° percentile but still under the maximum value or over the minimum value of the box plots (yellow outliers). More details can be observed in S7 Table in S1 File. Evaluating the parameters that the RP engine takes into account the creation of the plan's objectives, for the volume parameters a range of [0–88.3]% were flagged as outliers depending on the chosen model, [0–10]% as outliers outwith the 90° percentile or beneath the 10° percentile; for the PCS a range of [0–78.3]% were flagged as outliers depending on the chosen model, [0–18.3]% as outliers outwith the 90° percentile or beneath the 10° percentile (S8 Table in S1 File). These percentages are far higher than the same values observed in the internal validation.

It is even more interesting to discover that the totality of these outliers turned to good value for the specific OAR, or the 100% of the red and yellow outliers turned into a "green value", so that the value of the corresponding OAR fell in the right range of constraints after the optimization phase.

In perfect agreement with the internal validation, all KB-based plans were clinically acceptable in terms of PTVs coverage and OAR sparing. The PTV/OARs average dose-volume objectives were used to appraise the quality of the reference and RP dose distributions and were quantitatively analysed for PTV and OAR to investigate the differences. All DVHs parameters are listed in detail in Table 5.

About the 0.3% of 292 analysed dose-volume objectives in the clinical plans failed to reach the optimal constraints, while the respective RP plans succeeded. In 2 out of 10 cases

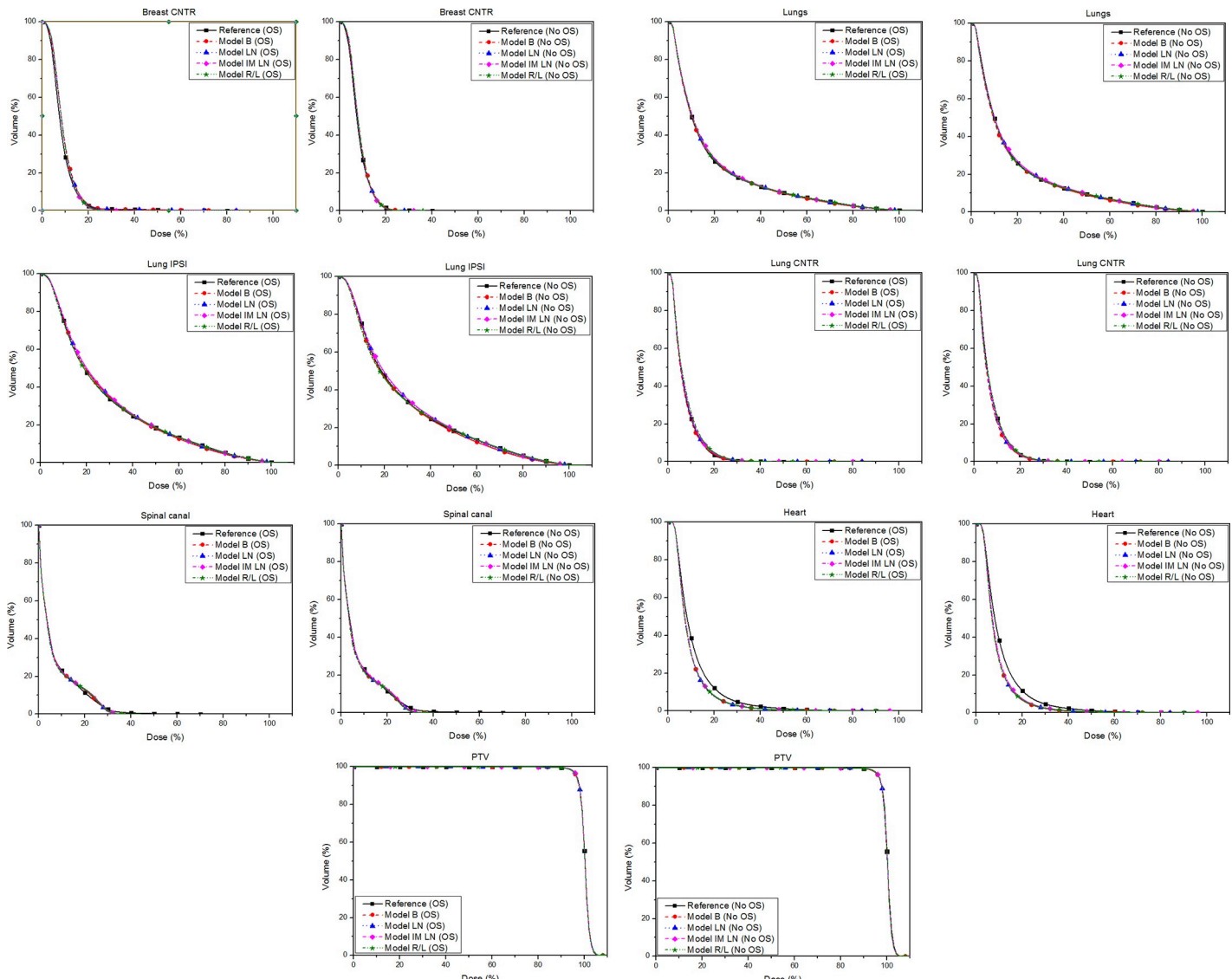

**Fig 1. Average DVHs for the main OARs and the PTV of the 30 patients of the internal validation set.** Comparison between Reference plan in black with the square symbol and model B in red with the round symbol, model LN in blue with the triangle symbol, model IM LN in magenta with the rhomboidal symbol and the sum of model R and L in green with the star symbol.

evaluation data set plans, 1 objective for each validated model, the RP plan failed at least one of the dose-volume constraints compared to the delivered plan in terms of PTV coverage, with a mean difference of less than 10%. Vice versa, when RP brings the constraints below the optimal value, the average difference is up to 20%, so happened for the $V_{5Gy}$ of the lungs of one analysed patient, that passed from the 61% in the reference plan to a 35% value using the B model (39% with the R model).

The external validation showed that RP based optimizations lead to systematic improvements in OAR sparing, with sometimes a poorer, but still clinically acceptable, PTV coverage. This sparing is greater than what observed in the internal validation. An average improvement of at least 16% is observed for the $V_{5Gy}$ of lungs in RP plans for both B and R models. The mean heart dose was almost halved with both models and alike the $V_{20Gy}$ for lung IPSI with

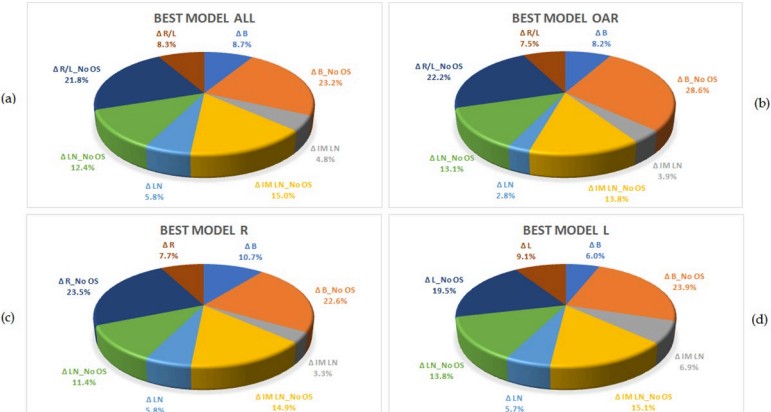

**Fig 2.** Pie charts show the difference Δ between reference and RP plans, or for every parameter listed in the above tables how many times the difference Δ is better for a model rather than another one. Pie chart (a) refers to the Δ of Table 1, pie chart (b) is weighted for the Δ of OARs only, pie chart (c) refers to the Δ of Table 2 for R treatments and pie chart (d) refers to the Δ of Table 2 for L treatments.

model B was decreased. As shown in Table 5, the models reduce the maximum dose for the spinal canal of more than 2 Gy on average. All these improvements are statistically significant, with p value often of the order of $10^{-5}$, highlighted with the (***) in Table 5, although, probably for the relatively small sample of the validation test, most of the data were not distributed normally and therefore, after the Shapiro Wilk test, we proceeded with the application of the Wilcoxon Sign rank test.

**Table 5. Overview and comparison of relevant parameters for both OARs and PTVs, averaged for the 10 patients of the external validation set treated on the right side, both for reference and model-based plans.**

| Model | Reference | | B_No OS | | | R_No OS | | |
|---|---|---|---|---|---|---|---|---|
| OAR/PTV | Average | Dev_st | Average | Dev_st | p value | Average | Dev_st | p value |
| **Breast CNTR** | | | | | | | | |
| $V_{10Gy}<5\%$ | 0.3 | ± 0.7 | 0.0 | ± 0.0 | | 0.0 | ± 0.1 | |
| **Lungs** | | | | | | | | |
| $V_{5Gy}<60\%$ | 46.1 | ± 8.7 | *29.5* | *± 8.1* | (***) | *31.2* | *± 6.5* | (***) |
| **Lung IPSI** | | | | | | | | |
| Dmean (Gy) | 13.0 | ± 2.0 | *8.6* | *± 2.0* | (***) | *9.7* | *± 1.8* | (***) |
| $V_{20Gy}<40\%$ | 21.8 | ± 5.2 | *11.4* | *± 4.3* | (***) | *14.4* | *± 3.9* | (***) |
| **Heart** | | | | | | | | |
| Dmean (Gy) | 5.0 | ± 1.5 | *2.9* | *± 0.3* | (***) | *2.6* | *± 0.4* | (***) |
| **Spinal canal** | | | | | | | | |
| Dmax (Gy) | 6.3 | ± 2.7 | *4.3* | *± 0.9* | (**) | *3.8* | *± 1.2* | (***) |
| **PTV Surg Bed** | | | | | | | | |
| Dmax (Gy) | 61.7 | ± 0.8 | *60.9* | *± 1.0* | (**) | *61.0* | *± 0.8* | (***) |
| $V_{98\%}$ | 85.2 | ± 4.4 | *87.9* | *± 5.5* | (**) | 87.1 | ± 4.7 | |
| $V_{95\%}>95\%$ | 98.6 | ± 1.4 | *97.2* | *± 2.6* | (**) | *97.8* | *± 1.8* | (**) |
| $V_{105\%}<5\%$ | 0.6 | ± 0.9 | *0.2* | *± 0.4* | (**) | *0.3* | *± 0.6* | (**) |
| **PTV RA** | | | | | | | | |
| $V_{95\%}>95\%$ | 97.7 | ± 2.1 | 97.0 | 2.4 | | 96.4 | ± 4.7 | |

P value is reported with *, ** or *** when significant as explained in the text.

## Discussions

The RP engine has been widely studied in recent years, applied on different sites: pelvis [60–62], esophagus [63], head and neck [64, 65], lung [66], spine [67] and brain [68]. Also breast site has already been investigated [47–49], but, to the best of our knowledge, not for VMAT treatments comprehensive of locoregional lymph nodes. These publications showed that the quality of KB plans, on average, outperformed that of the corresponding clinically accepted plans. The improvements observed in all the aforementioned studies were, partly, due to the use of the line optimization objectives defined slightly below the estimated DVH lower bound, i.e. the optimization is driven towards the best estimated DVH. DVHs of OARs can be estimated using RP models, trained using PCA and stepwise regression analysis.

The need arises for an evaluation of the model's performances, since the training set can be built in many ways and this has already been done for all the above listed sites. The results showed that plans generated with the assistance of RP exhibited improved dosimetric performance compared to the benchmark clinically accepted plans, however highlighting the need to identify properly outlier plans to better implement KB planning into the clinical practice.

Nevertheless, almost the totality of the outliers flagged at the beginning of the optimization process turned to good values for the specific OAR, or the value of the corresponding OAR fell in the right range of dose constraints after the optimization phase. This fact implies that even if a structure is considered "borderline" from the model statistic for its geometric or statistic aspects, at the end that structure can be properly optimized and it can enter the right range of acceptable values for the treatment plan.

Chang et al. showed that the performance of a RP to achieve dose constraints is still behind that of an experienced planner, and manual touch-up could be necessary, although RP based plans with a single optimization without any modifications could produce clinically acceptable plans [69]. The present results are indeed generated with no user interaction during the optimization run. In a few of the breast case studies, minor refinements are still required to smooth out small hotspots or to boost coverage in small regions. Patients where some compromise is required due to proximity or overlap of OARs with PTVs are likely to require further interactions and clinical decisions to compromise either coverage or OAR constraints, but initial optimization using the RP models gives an excellent starting point. For those RP plans that could not fulfill the plan acceptance criteria, minor manual touch-up was sufficient to make them clinically acceptable, with the same quality as those not requiring manual touch-up. Residual failures might be due to an insufficient predictive power of the model, which could be fixed by using greater training sample. In addition, there might be room to further refine the model with the information we have gathered in this study.

The "breast model" describes models applicable to breast cases, regardless the dose prescription, boost, whether simultaneous integrated or sequential, annexes or not lymph nodes, etc. The training set was determined without special selection criteria, if not that of feeding the models with previously treated plans is strongly desirable since these plans reflect treatment techniques and constraints that are clinically acceptable, complying our protocol. The guideline followed for the study was to include in the training set an adequate representation of the population to be sampled. The patient datasets were intentionally generated with a heterogeneous population in terms of tumor location, size and dose prescriptions (from 40.05 to 63.22 Gy, to a single volume or in SIB). However, as shown in this study, with judiciously chosen optimization objectives and a suitable training set, the challenges due to the trade-off between coverage needs and OARs tolerances can be overcome.

The RP plans are capable to meet stringent OAR constraints while still maintaining a good PTV coverage. The range of patient geometries in the model libraries still may not represent the full diversity of breast cancer cases due to individual differences. Therefore, special caution should

be taken when applying the model libraries to those patients whose geometry falls outside the range of the constituent plans in the libraries. Removing geometric outliers will reduce the variation of the anatomy within a model and thus resulting in more models necessary to cover all cases. For this reason Sheng et al [60] require further investigation about the idea of building one model that can predict equally well for all cases, to clear whether it is more advantageous to create a model on individual sites or on a combination of cases from some or all sites. This is what we attempted to do in our preliminary study. Our results reinforce the possibility of building effective broad-scope models and, likely, of suggesting that, to some extent, the use of heterogeneous datasets (in their geometric and dosimetric aspects) might be useful if not necessary.

Correlation between heterogeneity of the input data, the number of training cases needed, and generalization power of the models has been investigated generating different models for the different treatments listed above. We demonstrated that sub-KB models, developed by including only one type of treatment at the time (i.e. LN and IM LN models) have shown good performance, comparable but slightly worse than the general model. This could be due to the fact that these models include half of the plans of the general one, but also could be due to the very good power of generalization inherent RP, since the general model was trained with a cohort of mixed cases with equivalent incidence of all classes. A training set which samples the patient population with an adequate case mix can be used for a general purpose and it works better than the more specific model, as these preliminary findings testified. One special mention regards the splitting of right and left treatment sites, because the R and L models have shown as good performances as model B, being R model even better than B model. In particular, R model could be improved changing priorities, especially in heart objectives, because it has to be considered that, in this preliminary study, the priorities and the objectives are chosen to be the same for every model to better compare all the models.

Another important point regarded the impact of the dimension of the training set on the robustness of the KB model prediction. In fact, the number of patients used for training is a critical issue for the resulting quality of KB models. Cagni et al demonstrated the dependence of the DVH prediction performance on the size of the training patients in the model, which is more accurate with training sets $\geq$45 plans [23]. This result was in agreement with our RP models, consisting in at least 52-training plans. However, even with a training set of 114 patients, clinically relevant inaccuracies in predicted DVHs were observed, as well as we saw in our general model (B), trained with 120 plans.

Shubert et al performed a side cross-validation test to validate the usability of the same model irrespective of the beam energy selected for the plans [26] and no differences were observed between the plans optimized for 6 or 15 MV photons and all based on the same model. Huang et al appraised that a RP model configured with flattened high energy beams does not satisfy target dose coverage using un-flattened photons and may increase normal tissue exposure if applied to optimize lower energy beams [70]. In the clinical use of the models it has happened in cases of exceptionally large breasts to prefer the use of the 10 MV photons instead of the 6 MV and RP has been found to be effective also in these cases.

The creation of robust models was desirable by removing influential outliers while keeping plans that provide additional information, to create models that are exploitable to a potentially large number of users and, in this study, also to other institutions. The model validation was performed on two groups of cases and not used for the training; one set of cases from the clinic that train the models and one set from an external clinic, likewise to what investigated in some previous studies [24, 26, 27, 29–31]. Our results confirmed that the models, built with at least 52 patients from clinic 1, resulted adequate to properly optimize plans from clinic 2. The OARs sparing obtained using the models in clinic 2 is even higher than what reached in clinic 1. Among the other things, in clinic 2 the validation plans do not include the lymph nodes

irradiation, so the models created to work in more difficult cases, perform even better in simpler cases. The findings here reported confirm the possibility of using the models, generated and tested by clinic 1 in clinic 2, however, the adherence to the same guidelines could be facilitated and made stronger by the use of KB planning methods in a multicentric cooperative initiative [23]. Hence, RP may provide uniform plan quality across many centers. The multicentric validation demonstrated the possibility of sharing models among different institutes, highlighting the importance of an accurate validation for KB models [27].

The development and implementation of heart-sparing in breast RT techniques remains a priority. Breath-hold techniques [71–75] and VMAT [76] improve the heart sparing and decrease the risks of cardiac complication probabilities, in particular when the tumor bed is close to the heart or in the case of IM nodes irradiation. Decreases of heart $D_{mean}$ by more than 10% with KB plans were found in almost the totality of the plans, compared to an accepted average $D_{mean}$ value of 10 Gy. Darby et al. showed a linear increase in the relative rate of major coronary events with the heart $D_{mean}$, the excess relative risk per Gy is 7.4%/Gy [77]. Based on this, an average reduction of 1.44 Gy using the L model without optimization structures, which is the best obtained reduction for this parameter, could represent an approximately 10.7% decrease of the relative risk.

In the previous conventional optimization techniques, two resource-intensive processes were often required. Primarily, the creation of multiple "dummy" structures to aid the optimization process; for example, structures that considered overlap between OAR and PTV, ring structures to better conform the isodoses and optimization PTVs to help in PTV coverage. We proved that these structures are not necessary with RP through the creation of the two twin sets of models, with and without the optimization structures, demonstrating far better achievements with the latter models. Secondly, it was necessary to use a set of optimization objectives that required iteratively adjustment on a patient-by-patient basis, until a clinically acceptable plan could be achieved. With RP, these iterative steps are removed as the DVH objective generation is automatically tailored to each patient. With a suitable training set and an iteratively adjusted plan optimization parameter template, treatment plans achieved satisfactory DVH objectives even after a single optimization.

Planning time was not part of the study design, but some considerations are reported in the supporting information. In general, the increase in planning time, even if manual touch-up is needed to reach an optimum plan after RP optimization, is negligible compared with the total planning time for the manual plans.

Concerning the main aims of the study, the RP KBP approach was shown to be robust with respect to:

i.  The use of a general model to predict DVHs for whole breast with locoregional lymph nodes irradiation instead of a different model for each specific treatment

ii.  Single volume and SIB together

iii.  No optimization structures requested in the optimization phase

iv.  Good performances for the general model, comparable with its splitting into asymmetric models R and L

v.  Models created from one clinic generalizable to another one, showing optimal results.

## Conclusions

These are the first RP models that consider whole-breast irradiation comprehensive of nodal station with VMAT technique. All KB models used for planning allow a homogeneous plan

quality and some dosimetric gains, as we saw in both internal and external validation. The results here presented support the conclusion that KB planning systems can improve the mean plan quality of a single institution and of other institutions that have similar protocols. Sub-KB models, developed by splitting right and left breast cases or including only whole breast with locoregional lymph nodes, have shown good performance but slightly worse than the general model. Finally, models generated without the optimization structures, performed better than the original ones. This KB approach effectively refines plans optimization and could be helpful in clinical practice, which can reduce the dependence of plan quality on planner skills thus increasing the robustness and homogeneity of the radiotherapy process. This can also be regarded as powerful tools for knowledge sharing and early education, in particular in a center where there are planners with different expertise, in order to standardize and improve the quality of the plans. The external validation of the model by another center appraise the power of RP generalization, although this can be considered as one of the many feasibility studies which contribute to the veracity of this statement. Further external validations of the model by other centers would definitely certify the robustness of the proposed RP models' power.

## Supporting information

**S1 File. [78].**
(DOCX)

**S1 Fig. Average DVHs for the main OARs and the PTV of the 18 patients of the internal validation set who were treated on right side.** Comparison between Reference R plans in black with the square symbol and model R plans in green with the star symbol.
(TIF)

**S2 Fig. Average DVHs for the main OARs and the PTV of the 12 patients of the validation set who were treated on the left side.** Comparison between Reference L plans in black with the square symbol and model L plans in green with the star symbol.
(TIF)

**S3 Fig. Average DVHs for the main OARs and the PTVs of the 12 patients of the validation set who were treated in SIB.** Comparison between Reference SIB plans in black with the square symbol and model B (SIB plans only) in red with the round symbol, model LN (SIB plans only) in blue with the triangle symbol, model IM LN (SIB plans only) in magenta with the rhomboidal symbol, the sum of model R and L (SIB plans only) in green with the star symbol and model SIB in cyan with the cross symbol.
(TIF)

## Author Contributions

**Conceptualization:** Lorenzo Placidi, Mattia Polsoni, Giulia Rambaldi.

**Data curation:** Maria Rago, Lorenzo Placidi.

**Formal analysis:** Maria Rago.

**Investigation:** Maria Rago.

**Methodology:** Maria Rago, Lorenzo Placidi, Mattia Polsoni.

**Project administration:** Vincenzo Valentini, Marco De Spirito.

**Software:** Maria Rago.

**Supervision:** Lorenzo Placidi, Luigi Azario.

**Validation:** Maria Rago, Mattia Polsoni, Giulia Rambaldi.

**Visualization:** Davide Cusumano, Francesca Greco, Luca Indovina, Sebastiano Menna, Elisa Placidi, Gerardina Stimato, Stefania Teodoli, Gian Carlo Mattiucci, Silvia Chiesa, Fabio Marazzi, Valeria Masiello.

**Writing – original draft:** Maria Rago.

**Writing – review & editing:** Lorenzo Placidi, Giulia Rambaldi, Sebastiano Menna, Elisa Placidi, Fabio Marazzi, Valeria Masiello.

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
