## [Decision Letter · Decision Letter 0]

5 Nov 2020

PONE-D-20-26449

Evaluation of a generalized knowledge-based planning performance for VMAT irradiation of breast and locoregional lymph nodes - internal mammary and/or supraclavicular regions

PLOS ONE

Dear Dr. Placidi,

Thank you for submitting your manuscript to PLOS ONE. After careful consideration, we feel that it has merit but does not fully meet PLOS ONE’s publication criteria as it currently stands. Therefore, we invite you to submit a revised version of the manuscript that addresses the points raised during the review process.

We look forward to receiving your revised manuscript.

Kind regards,

Dandan Zheng, PhD

Academic Editor

PLOS ONE

Journal Requirements:

2.) Please amend your list of authors on the manuscript to ensure that each author is linked to an affiliation. Authors’ affiliations should reflect the institution where the work was done (if authors moved subsequently, you can also list the new affiliation stating “current affiliation:….” as necessary).

Reviewers' comments:

Reviewer's Responses to Questions

**Comments to the Author**

1. Is the manuscript technically sound, and do the data support the conclusions?

Reviewer #1: Partly

Reviewer #2: Yes

2. Has the statistical analysis been performed appropriately and rigorously? 

Reviewer #1: No

Reviewer #2: Yes

3. Have the authors made all data underlying the findings in their manuscript fully available?

Reviewer #1: Yes

Reviewer #2: Yes

4. Is the manuscript presented in an intelligible fashion and written in standard English?

Reviewer #1: Yes

Reviewer #2: Yes

5. Review Comments to the Author

Reviewer #1: The manuscript was not labeled by line numbers, which makes it hard to make specific comments/recommendations. I will try my best.

First, I would like to point out that the validation sample size from the external institution (10 patients) is too small to draw any conclusion statistically. It’s better to demonstrate the results as a feasibility study rather than conclude the model can be used as a powerful tool across institutions without more comprehensive study (end of the conclusion).

Second, there are multiple places where the improvement of cord doses were mentioned. As the author has already noticed, this result does not have statistical significance. In another word, if you rebuild the model with a subset of the patient, or apply to a different validation group, likely this result will not hold. I would recommend the author to remove these comments about cord dose.

Another issue I have noticed is that the PTV coverage seems to be decreasing in the plans generated from KB models. This is a potential bias when comparing the OAR doses. Although I don’t believe it will significantly impact the dose for volumetric OARs, it may make the difference less pronounced. I would recommend the author to renormalize the reference plan and model-generated plan to the same coverage before comparison.

In the results session, what’s the reason that some results (statistically significant & clinically relevant) from the table 1 are ignored? For example, there seems to be dosimetric impacts of lung V20 and V10 volumes from the data, which is commonly a trade off between heart dose, but not mentioned in the results session. In general, a few things need to be summarized in the results: Which OARs do not show statistical significance; Which OARs do show statistical significance, and whether they are clinically relevant (For example the increase of max dose in lung may not be a concern but this result needs to be mentioned in the results session)

Regarding table 1, why is the table split into two parts? Please explain this explicitly in the title of the table.

I wouldn’t call 3DCRT extremely time consuming because those plans are simpler in the planning and treatment process compared with IMRT plans. Unless solid reference is provided I would suggest the author to remove this statement.

Reviewer #2: This manuscript is a good compliment to previously published works by applying RP to VMAT with nodes.

In Methods and Materials, 1st paragraph, last sentence. Change plan to plans.

What time period did the database include? The use of internal and external validation is very helpful in appreciating the potential to use RP in a clinical setting.

6. PLOS authors have the option to publish the peer review history of their article (what does this mean?). If published, this will include your full peer review and any attached files.

Reviewer #1: No

Reviewer #2: No

---

## [Author Response · Author response to Decision Letter 0]

30 Nov 2020

Dear Editor-in-Chief,

thank you for giving us the opportunity to submit a revised draft of the manuscript titled “Evaluation of a generalized knowledge-based planning performance for VMAT irradiation of breast and locoregional lymph nodes – internal mammary and/or supraclavicular regions” by Maria Rago et al. We really appreciated the time and effort that you and the reviewers have dedicated to provide us your valuable feedback of the paper. We have been able to incorporate changes to reflect most of the suggestions provided by the reviewers and we have modified the paper also providing the required information. 

The following are our responses to the reviewers’ comments, labelled according to the order indicated.

Reviewer #1: 

The manuscript was not labeled by line numbers, which makes it hard to make specific comments/recommendations. I will try my best.

We do apologize for the inconvenience. In the revised unmarked, as well as in the revised with track changes manuscript we have included the line numbers. 

Comment 1. First, I would like to point out that the validation sample size from the external institution (10 patients) is too small to draw any conclusion statistically. It’s better to demonstrate the results as a feasibility study rather than conclude the model can be used as a powerful tool across institutions without more comprehensive study (end of the conclusion).

Thank you for pointing this out. We agree with the reviewer that the sample size from the external institution (10 patients) is relatively small. Nevertheless, bringing the models to validate and obtain excellent results even outside the training hospital is part of a series of feasibility studies thanks to which is possible to conclude the power of RapidPlan model. Internal validation has been performed on 30 patients and therefore we decided (also based on the availability of the external institution to validate our model) to set to 10 the patients for the external validation. Additionally, in most of the scientific publications related to KB models with RapidPlan and more in general with auto-planning models, single external validation samples size is comparable with the one employed in our study. Below are reported some references and the relative number of the sample size used in the external validation:

- Kamima et al. Multi-institutional evaluation of knowledge-based planning performance of volumetric modulated arc therapy (VMAT) for head and neck cancer Physica Medica 64 (2019) 174–181 https://doi.org/10.1016/j.ejmp.2019.07.004

o Type of auto-planning: RP

o 2 cases for each center (5 centers in total)

- Schubert et al. (2017). Intercenter validation of a knowledge based model for automated planning of volumetric modulated arc therapy for prostate cancer. The experience of the German RapidPlan Consortium. PLOS ONE. 12. e0178034. 10.1371/journal.pone.0178034.

o RP

o 10, 7, 6, 7, 13, 10, 7 cases for each center respectively (5 centers in total)

- Fogliata et al. (2015) Performance of a Knowledge-Based Model for Optimization of Volumetric Modulated Arc Therapy Plans for Single and Bilateral Breast Irradiation. PLoS ONE 10(12): e0145137. doi:10.1371/journal.pone.0145137

o Type of auto-planning: RP

o 15 and 10 patients for each center respectively (2 centers in total)

- Fogliata et al. A broad scope knowledge based model for optimization of VMAT in esophageal cancer: validation and assessment of plan quality among different treatment centers Radiation Oncology (2015) 10:220 DOI 10.1186/s13014-015-0530-5

o Type of auto-planning: RP

o 10 patients (1 center)

- Roach et al. Adapting automated treatment planning configurations across international centres for prostate radiotherapy Physics and Imaging in Radiation Oncology 10 (2019) 7–13 https://doi.org/10.1016/j.phro.2019.04.007

o Type of auto-planning: AP Pinnacle

o 10 patients for each center (1 center)

Thus, multi external institutions validation could definitely support the conclusion that the evaluated model can be used as a powerful tool across institutions. For this reason, we have modified the text of the manuscript highlighting how the results of the single external institution validation support the feasibility of this study of employ the proposed RP model in external institution. Indeed, we strongly hope that our work could be further validated to potential RapidPlan users in our community, increasing the validation sample size. 

As suggested by the reviewer, we have point out this important issue in the new version of the manuscript, highlighting in the conclusion section as reported below: 

The external validation of the model by another center appraise the power of RP generalization, although this can be considered as one of the many feasibility studies which contribute to the veracity of this statement. Further external validations of the model by other centers would definitely certify the robustness of the proposed RP models’ power.

Comment 2. Second, there are multiple places where the improvement of cord doses were mentioned. As the author has already noticed, this result does not have statistical significance. In another word, if you rebuild the model with a subset of the patient, or apply to a different validation group, likely this result will not hold. I would recommend the author to remove these comments about cord dose.

We thank the reviewer for this remark. As suggested, we have removed these comments about the cord dose. 

Comment 3. Another issue I have noticed is that the PTV coverage seems to be decreasing in the plans generated from KB models. This is a potential bias when comparing the OAR doses. Although I don’t believe it will significantly impact the dose for volumetric OARs, it may make the difference less pronounced. I would recommend the author to renormalize the reference plan and model-generated plan to the same coverage before comparison.

We thank the reviewer to highlight this relevant issue. Based on the results in table 1 (now table 1,2, 3 and 4) PTV coverage slightly decreasing in the plans generate from the KB models. We agree that this is a potential bias when comparing OARs doses: obviously, decreasing the target coverage could result in a better OARs sparing and therefore the comparison between the reference and the model-generated plan would not be fair. However, we were comfortable in compare these results since the obtain PTV coverage in the model-generated plans was always within our clinical requirement (PTV V95%>95%). Moreover, the PTV coverage variation between the reference plan and the model-generated plan is small if considered table 2 and 4 were the mean, minimum and maximum variation were 0.4%/0.1%/0.9% and 0.4%/0%/1.2% respectively. A strict and right comparison would have required to re-normalize the reference plan and model-generated plan (as suggested by the reviewer) to the same coverage before comparison. Nevertheless, both plans have been normalized to the targe mean and we do not expect a significantly impact on the volumetric OARs. In addition, this is the normalization scheme implemented in our clinical practice and therefore the introduction of a new normalization scheme would have not allowed to use in the clinical practice the evaluated RP model. Even if a new normalization scheme would have been applied to the reference and model-generated plans, we do not expect a difference so relevant as to change the conclusions of the study, which should also be rebuilt from scratch. 

Comment 4. In the results session, what’s the reason that some results (statistically significant & clinically relevant) from the table 1 are ignored? For example, there seems to be dosimetric impacts of lung V20 and V10 volumes from the data, which is commonly a trade off between heart dose, but not mentioned in the results session. In general, a few things need to be summarized in the results: Which OARs do not show statistical significance; Which OARs do show statistical significance, and whether they are clinically relevant (For example the increase of max dose in lung may not be a concern but this result needs to be mentioned in the results session)

We thank the reviewer to pointing this out. As suggested, we have better described in the text of the manuscript of the results of table 1 (now table 1,2,3 and 4). In particular we have highlighted the following point: which OARs do not show statistical significance; which OARs do show statistical significance and whether they are clinically relevant. These edits are shown here below: 

Moreover, the maximum dose for the spinal canal is about 1 Gy less on average, although not statistically significant. Small improvements are observed for the V5Gy and the Dmean of lungs in RP plans, although they are significant only for the B model. 

Some OARs shown slight dose increases, such as the contralateral breast or the LADCA, for LN and IM LN models. The same parameters turn to be slightly but statistically improved in model B, for example the V10Gy for the contralateral breast or the Dmax for the LADCA. Almost no statistically significant results were observed for lung IPSI, spinal canal, oesophagus or thyroid.

Comment 5. Regarding table 1, why is the table split into two parts? Please explain this explicitly in the title of the table.

We thank the reviewer to highlight this issue. In order to improve the readability of the results, we have better organize table 1, as well as table 2, in four different tables (table 1,2,3 and 4), sorting not only for the validation data set, but also for OARs and PTV. The titles of the tables have been also improved to better describe and explain these results. We have subsequentially re-numbered all the tables in the manuscript, as well as in the supporting information. 

Comment 6. I wouldn’t call 3DCRT extremely time consuming because those plans are simpler in the planning and treatment process compared with IMRT plans. Unless solid reference is provided I would suggest the author to remove this statement.

We agree that this statement can be misunderstanding, especially without a solid reference to support it. We have therefore removed this statement and replace it highlighting how 3DCRT plan can strongly planner dependent, as following: 

Moreover, 3DCRT plans can be very operator dependent.

Reviewer #2 

This manuscript is a good compliment to previously published works by applying RP to VMAT with nodes.

We thank the reviewer for the valuable feedback received and we are glad she/he has appreciated our manuscript, especially in providing to our community a missing information relate to the application of RP to VMAT with nodes. 

Comment 1. In Methods and Materials, 1st paragraph, last sentence. Change plan to plans.

We thank the reviewer for this remark. We have modified the text of the manuscript as suggested: 

Each model was built on a range of plans from 52 to 120 (depending on the model).

Comment 2. What time period did the database include? The use of internal and external validation is very helpful in appreciating the potential to use RP in a clinical setting.

Thank you for pointing this out. The selected time period of the database was January 2017 to September 2019. The text of the manuscript has been modified as following:

A set of clinical plans elaborated from January 2017 to September 2019 was included in this retrospective study. 

We totally agree with the reviewer that the use of internal and external validation is a extremely important step to demonstrate the potential use of RP in a clinical setting.

---

## [Decision Letter · Decision Letter 1]

26 Dec 2020

Evaluation of a generalized knowledge-based planning performance for VMAT irradiation of breast and locoregional lymph nodes - internal mammary and/or supraclavicular regions

PONE-D-20-26449R1

Dear Dr. Placidi,

We’re pleased to inform you that your manuscript has been judged scientifically suitable for publication and will be formally accepted for publication once it meets all outstanding technical requirements.

Kind regards,

Dandan Zheng, PhD

Academic Editor

PLOS ONE

Additional Editor Comments (optional):

Reviewers' comments:

Reviewer's Responses to Questions

**Comments to the Author**

1. If the authors have adequately addressed your comments raised in a previous round of review and you feel that this manuscript is now acceptable for publication, you may indicate that here to bypass the “Comments to the Author” section, enter your conflict of interest statement in the “Confidential to Editor” section, and submit your "Accept" recommendation.

Reviewer #1: All comments have been addressed

Reviewer #2: All comments have been addressed

2. Is the manuscript technically sound, and do the data support the conclusions?

Reviewer #1: Yes

Reviewer #2: Yes

3. Has the statistical analysis been performed appropriately and rigorously? 

Reviewer #1: Yes

Reviewer #2: Yes

4. Have the authors made all data underlying the findings in their manuscript fully available?

Reviewer #1: Yes

Reviewer #2: Yes

5. Is the manuscript presented in an intelligible fashion and written in standard English?

Reviewer #1: Yes

Reviewer #2: Yes

6. Review Comments to the Author

Reviewer #1: (No Response)

Reviewer #2: The author fully addressed the issues brought up in the previous review. The investigation into using RapidPlan in the clinic shows the potential in future RT planning versus the current clinical workflow.

7. PLOS authors have the option to publish the peer review history of their article (what does this mean?). If published, this will include your full peer review and any attached files.

Reviewer #1: No

Reviewer #2: No

---

## [Editor Report · Acceptance letter]

30 Dec 2020

PONE-D-20-26449R1 

Evaluation of a generalized knowledge-based planning performance for VMAT irradiation of breast and locoregional lymph nodes - internal mammary and/or supraclavicular regions 

Dear Dr. Placidi:

I'm pleased to inform you that your manuscript has been deemed suitable for publication in PLOS ONE. Congratulations! Your manuscript is now with our production department. 

Kind regards, 

on behalf of

Dr. Dandan Zheng 

Academic Editor

PLOS ONE